# Efficient Computation of Spatial Entropy Measures

**DOI:** 10.3390/e25121634

**Published:** 2023-12-08

**Authors:** Linda Altieri, Daniela Cocchi, Giulia Roli

**Affiliations:** Department of Statistical Sciences, University of Bologna, 40126 Bologna, Italy; daniela.cocchi@unibo.it (D.C.); g.roli@unibo.it (G.R.)

**Keywords:** Batty’s entropy, biodiversity, distance-based entropy, gorilla nesting sites, Leibovici’s entropy, multinomial data, O’Neill’s entropy, point data

## Abstract

Entropy indices are commonly used to evaluate the heterogeneity of spatially arranged data by exploiting various approaches capable of including spatial information. Unfortunately, in practical studies, difficulties can arise regarding both the availability of computational tools for fast and easy implementation of these indices and guidelines supporting the correct interpretation of the results. The present work addresses such issues for the most known spatial entropy measures: the approach based on area partitions, the one based on distances between observations, and the decomposable spatial entropy. The newly released version of the R package SpatEntropy is introduced here and we show how it properly supports researchers in real case studies. This work also answers practical questions about the spatial distribution of nesting sites of an endangered species of gorillas in Cameroon. Such data present computational challenges, as they are marked points in continuous space over an irregularly shaped region, and covariates are available. Several aspects of the spatial heterogeneity of the nesting sites are addressed, using both the original point data and a discretised pixel dataset. We show how the diversity of the nesting habits is related to the environmental covariates, while seemingly not affected by the interpoint distances. The issue of scale dependence of the spatial measures is also discussed over these data. A motivating example shows the power of the SpatEntropy package, which allows for the derivation of results in seconds or minutes with minimum effort by users with basic programming abilities, confirming that spatial entropy indices are proper measures of diversity.

## 1. Introduction

The interest in diversity measures spans a variety of fields of applications, such as ecological and biodiversity studies [1]. Over the years, these studies have focused on heterogeneity indices based on Shannon’s entropy [2]. The original version of Shannon’s entropy only considers the probabilities (or their estimates) of the categories of the variable under study. In many real cases, e.g., environmental applications, this can be a major limitation since often, external factors (such as covariates and spatial effects) can affect the occurrence of a specific category. The knowledge of structures, such as spatial dependence, may help in understanding the distribution of categories across observations. The idea is that a dataset with randomly scattered observations should have a higher entropy than one with observations arranged according to a spatial structure since the spatial information should decrease the surprise. Unfortunately, Shannon’s entropy fails to capture such differences. These aspects are well known and widely discussed in the literature (see, e.g., [3]). Following this idea, several proposals have been made for the inclusion of information about the spatial structure of data in entropy measures for both discrete and continuous space. Space may be considered, for instance, by building a meaningful partition of the observation window into sub-areas and studying the evenness of the distribution of the phenomenon of interest across areas [4]. This can be done with or without considering a neighbourhood structure among areas, which introduces a system of distances and rankings [5]. Another proposal is based on the simultaneous consideration of couples or sets of observations based on a chosen distance, commonly called co-occurrences, and on studying the heterogeneity among such sets [6,7,8]. Each approach has specific properties, with both advantages and drawbacks when applied in studies. In practical work, a main obstacle to the usage of spatial entropy measures is the lack of a proper computational tool that could be exploited by applied scientists, such as biologists and ecologists, to feasibly obtain results and support the interpretation of the phenomenon under study. There is a need to properly use and interpret the different measures as diversity indices for spatial data. One further challenging aspect when dealing with spatial measures is scale dependence. This has been widely discussed by several authors (see, e.g., [9,10]), is closely related to the well-known modifiable areal unit problem (MAUP) [11], and especially affects studies on areal and grid data, such as in urban spatial analysis, where different choices of the spatial pattern can be adopted and the results are often compared among various cities. As a consequence, spatial entropy results may vary according to scale measurements and spatial patterns. Several authors have proposed incorporating the fractal dimension approach into spatial entropy measurements to properly control the complexity of a given pattern and ultimately achieve scale-free indices [9,12,13,14].

The present paper contributes to the field of entropy indices for spatial data in three ways. The first contribution is the introduction of a newly released version of a specific package of functions for the R software (version ≥ 4.0), named SpatEntropy. The package significantly contributes to the use of diversity indices in applied studies since no other R package covers the topic of spatial entropy measures. The package is free to download and handles the heavy lifting, allowing easy accessibility and interpretation of the results even for scientists with very basic programming abilities. In the present paper, we provide the R code lines for reproducing all the main spatial entropy measures, along with practical software details. Further details for the beginner user are given in Appendix B, and the computational time for the main package functions is reported in Table 6. A second contribution consists of providing a unified framework for the different approaches to spatial entropy measures available in the literature. Indices are discussed and compared, and interpretation guidelines are offered in order to help researchers make the best choice for the specific case study. The third contribution of the paper is addressing a computationally challenging dataset that contains the nesting sites of an endangered species of gorillas in the Kagwene Gorilla Sanctuary in Cameroon [15]. This motivating dataset consists of a point pattern in continuous space over an irregularly shaped observation area, with several marks constituting additional information about the nesting sites, along with both qualitative and quantitative environmental variables. The dataset stimulates a lot of possible research directions with different levels of complexity: the researcher may consider information on the nesting sites or the spatial covariates, space may need to be discretised, and the observation area may need to be modified to a more regular shape depending on the available computational tools. We address some of these challenges from several perspectives, showing the different contributions of space and guiding the reader in making choices and properly interpreting the corresponding results. Since in the present application, the spatial pattern of the database under study is uniquely defined and fixed by the institute that collected the data, when considering the original point data, the issue of scale dependence is not relevant, whereas it should be accounted for when choosing the grid resolution for space discretisation. Therefore, in order to highlight this crucial aspect, an empirical sensitivity study assessing the effect of different choices for grid resolution on spatial entropy is carried out and discussed in the present work.

The paper is organised as follows. The remainder of the present section gives the necessary preliminary information: the data example is described, and the new release of the SpatEntropy package is introduced. In Section 2, we offer some essential background information and show how to implement and interpret Shannon’s entropy. Section 3, Section 4 and Section 5 present the three main approaches to spatial entropy measures: each section starts with a theory, formally defining the indices, and contains a practical subsection with computations and another subsection with an interpretation of the results on the gorilla dataset. Where applicable, an additional subsection discusses the consequences of the spatial scale. The focus of this paper is on practical and computational aspects. For this reason, only the essential theoretical information is given, and appropriate references for deepening the theory are provided. An overall discussion is presented in Section 6, and the concluding remarks are presented in Section 7. Appendix B contains a detailed guide for the reader on the initial steps for practical work using R and the SpatEntropy package.

### 1.1. Data Presentation

The data considered in this work come from a study by the Wildlife Conservation Society’s Takamanda-Mone Landscape Project (https://cameroon.wcs.org, accessed on 10 January 2023) on the nesting habits of gorillas in the Kagwene Gorilla Sanctuary, an area in Cameroon. The dataset is a point pattern, i.e., a collection of points giving the exact spatial locations of n=647 nesting sites of gorilla groups observed in the area of interest. The spatial location is given in the Universal Transverse Mercator (UTM) coordinate system (Zone 32N) and expressed in metres. The observation area *T*, also known as the observation window in standard point process terminology, is an irregular polygon marking the boundary of the Kagwene Gorilla Sanctuary. Further details about the data collection are reported in [15]. The dataset is very rich; our working example will employ part of the available information.

Mark variables, i.e., additional information about the nesting sites, are attached to the points. We only focus on the mark *group*, a binary variable with values ‘major’ and ‘minor’, identifying the size of the gorilla group that constructed the nesting site. Other marks focusing on temporal aspects are provided with the data.

Spatial covariates are also available over the study region:*Elevation* of terrain, a continuous variable expressed in metres;*Vegetation* or cover type, a categorical variable with the values ‘Disturbed’ (highly disturbed forest), ‘Grassland’ (savannah), ‘Primary’ (primary forest), and ‘Secondary’ (secondary forest);*Waterdist*, a continuous variable for the Euclidean distance from the nearest water body expressed in metres.

The data and environmental covariates are shown in Figure 1. More details about the data structure can be found in Section 1.3.

The data are publicly available as part of the spatstat package in the R software.

### 1.2. Introducing the New Release of SpatEntropy

The R package SpatEntropy was first released in 2018 with the aim of providing a computational framework for the main existing spatial entropy measures. It provides an original contribution to the R community, as there is currently no other package available for such indices. Since it is meant for maximum diffusion among applied scientists, and not only for statisticians or programming experts, it has been gradually improved to become more user-friendly and fairly easy to approach. The initial version is presented in [16]. Subsequently, there have been a number of updates, with relevant changes and improvements.

The present work refers to the package version SpatEntropy 2.2-4, released in November 2023 (https://CRAN.R-project.org/package=SpatEntropy). The package works with both grid and point data. It supports both regular or irregularly shaped observation areas of any size and based on any coordinate system. For discrete data arranged on grids (raster/pixel data), cells are allowed to be rectangular. Point data may or may not be marked by additional variables, which become the categories for the computation of the entropy indices. Computational issues due to small areas in one of the entropy indices (see Section 3) are now automatically dealt with. Functions for the various entropy measures are now similar regarding input arguments and output structure. The output has been extended to return the minimum and maximum values of each index for easy interpretation. Moreover, in the latest version, all functions automatically return useful plots for effective explanation and dissemination of the results. The computational efficiency has been improved, and the results are now available in seconds or minutes, even for large datasets. Table 6 in Section 6 reports the computational times for all entropy indices, measured on a 2019 Windows Surface Pro 6 with an i7-8650U processor. The computational times are recorded using the R package microbenchmark: each function is run 1000 times, and the means and interquartile differences (*IQDs*) of the computational times are computed.

In the subsequent sections, we report in detail the main code for following the steps in this paper. We give a general interpretation of the output of the main entropy indices, but we do not give technical details about the produced objects in R, as we do not aim to produce a purely software-oriented user manual. Each function has a detailed help page in R, and further information can be found in [16].

### 1.3. Computations for the Motivating Example

In the following, we assume that the reader is able to start working with R and has downloaded the SpatEntropy package (if not, please refer to Appendix B):

  > library(SpatEntropy); library(spatstat).The following command line

  > gorillasshows preliminary information about the data object: the number of observed points is 647, where each point represents a nesting site of a gorilla group; three mark variables (*group*, *season*, *date*) have been collected, and the enclosing rectangle of the area is [580,440.4, 585,998.8] × [674,156.5, 678,732.2] m. The polygonal area of the sanctuary, enclosed within the rectangle, has a size of |T|=19.874 square kilometres.

For the entropy functions, we need to store a few different versions of the original dataset. First, to save and plot a version of the dataset without the marks, you can write


  > ungorillas = unmark(gorillas)


  > plot.ppp(ungorillas, pch = 16, main = "Gorilla nesting sites")which produces a similar plot to that in the upper-left panel of Figure 1.

Second, a discretised version of the data over a pixel grid is needed. The chosen grid resolution follows the grid resolution of the covariates (this choice is discussed later in this paper). Covariates are available as 149 × 181 = 26,969 pixel images:


  > elev = gorillas.extra$elevation



  > water = gorillas.extra$waterdist


  > veget = gorillas.extra$vegetation
They can be plotted with commands like

  > plot(elev, col = terrain.colors(100))
which yields the plot in the upper-right panel of Figure 1. The soil elevation ranges from around 1100 to around 2100 m, whereas the water distance ranges from 0 to 418 m. A summary of the preliminary information is shown in Figure 2.

To discretise data, the following lines of command can be used


  > discrgorillas = pixellate(ungorillas, W = gorillas$window,



      dimyx = elev$dim)


  > discrgorillas$v[discrgorillas$v >= 1] = 1These create a binary variable $X^{\left(discr\right)}$ with the values ‘nest’and ‘no nest’ for each pixel. The notation (discr) stands for *discrete* and is introduced to discern pixel grids from point pattern data. Inside the polygonal border, there are $n_{1}^{\left(discr\right)}=549$ pixels with at least one nesting site, and $n_{0}^{\left(discr\right)}=$ 20,513 pixels with no nesting sites. This grid approximation leads to a reduction in the recorded nesting sites with respect to the original number of n=647 sites, as some pixels contain more than one nest. The consequences of the grid choice are analysed throughout the paper, with a final discussion in Section 6. Inside the polygonal boundary, the number of pixels is $n^{\left(discr\right)}=549+$ 20,513 = 21,062. A total of 5907 pixels is classified as NA, i.e., *not available* missing data, as they lie outside the polygonal border of the window. The functions of the SpatEntropy package are able to discard the NA pixels without affecting the computations. A grayscale plot of the discrete dataset is shown in the right panel of Figure 3.

The last version of the dataset only considers the mark *group* and is built as follows:


  > gorillasgroup = gorillas



  > marks(gorillasgroup) = marks(gorillas)[ , 1]


  > plot.ppp(gorillasgroup, pch = 19, cols = 1:2)
where the last line produces the plot displayed in the left panel of Figure 3. Information about the group’s absolute and relative frequencies is discussed in the next subsection.

## 2. The Basics of Entropy for Environmental Data

The first diversity measure based on entropy, which is popular in environmental applications, was proposed by Shannon [2] and is thus commonly called Shannon’s entropy. Given a categorical variable *X* with *I* possible outcomes (or categories), Shannon’s entropy is defined as:(1)$$H\left(X\right)=\sum_{i=1}^{I}p\left(x_{i}\right)\log\frac{1}{p(x_{i})},$$
where $p(x_{i})$ for i=1,⋯,I is the probability of occurrence of category *i*. The index H(X) is non-negative and in the range of [0,logI] and expresses the average amount of heterogeneity, or diversity, across observations of a variable *X*. When the probabilities are (almost) equal, the entropy approaches its maximum, as it is hard to predict the category of the next outcome, and its observation carries the maximum amount of surprise. When one or a few categories are predominant, it is easier to guess the category of an unobserved outcome, and the entropy decreases.

Shannon’s entropy can be computed over both grid and point data, provided that they present at least two categories. The probabilities $p(x_{i})$ are usually unknown and estimated through the data relative frequency $n_{i}/n$, i.e., the number of observations presenting category *i* over the total number of observations. A discussion on the appropriateness of such a choice for $p(x_{i})$, as well as more sophisticated alternatives, can be found in [17]. The data relative frequency is an appropriate choice when the goal of the study is to describe the data, as is the case in the present work, rather than make predictions.

### 2.1. Computations on the Motivating Example

We first compute Shannon’s entropy on the point pattern, based on the mark *group* (Figure 3, left panel). In the continuous space, the binary variable $X^{\left(cont\right)}$ is considered with the two categories $x_{1}^{\left(cont\right)}$ = ‘minor’ and $x_{0}^{\left(cont\right)}=$ ‘major’. The function runs as

  > shannon(gorillasgroup)The output shows the absolute value for the entropy, i.e., $H(X^{\left(cont\right)})=0.69$, then it displays its range [0,log2=0.69], and returns the relative entropy $H_{rel}\left(X^{\left(cont\right)}\right)=0.995$. A table follows in the output containing details about the absolute and relative frequencies of each category to help the researcher investigate the results, as carried out in the next subsection. The output information is summarised in the left-hand side of Table 1.

Shannon’s entropy can be also computed for discrete data over the binary variable $X^{\left(discr\right)}$ with the categories $x_{1}^{\left(discr\right)}$ = ‘nest’ and $x_{0}^{\left(discr\right)}=$ ‘no nest’:

 > shannon(discrgorillas$v)resulting in the same output structure as above. In this case, the entropy value is $H(X^{\left(discr\right)})=0.12$, with $H_{rel}\left(X^{\left(discr\right)}\right)=H\left(X^{\left(discr\right)}\right)/\log2=0.17$ (details given in the right-hand side of Table 1).

### 2.2. Interpretation of Results

For point data, Shannon’s relative entropy $H_{rel}\left(X^{\left(cont\right)}\right)=0.995$ approaches its theoretical maximum, expressing a very high level of diversity of the gorilla group size. The left-hand side of Table 1 shows that the probabilities of the two group sizes are similar: $p(x_{0}^{\left(cont\right)})=0.54$ and $p(x_{1}^{\left(cont\right)})=0.46$; therefore, it is hard to predict whether a nesting site is built by a minor or major group, and the information carried by observing the outcomes is high. Unfortunately, such an entropy is invariant with respect to the spatial distribution of the nesting sites.

For grid data, the results are completely different: the relative entropy $H_{rel}\left(X^{\left(discr\right)}\right)=0.17$ is 17% of its theoretical maximum. Such an entropy is measured over the categories ‘nest’ and ‘no nest’. The value indicates that the heterogeneity (or diversity) of the system is very low, i.e., that observing outcomes does not carry much information, as they will very likely be 0-valued, i.e., ‘no nest’, pixels. Indeed, by looking at the frequencies on the right-hand side of Table 1, the probability (estimated through the relative frequency) $p(x_{0}^{\left(discr\right)})=0.97$ is largely predominant. Note that such values depend on the grid choice and finer grids would make the entropy tend to 0 because $p(x_{0}^{\left(discr\right)})$ approaches 1; this is shown in the next subsection. We must remark that Shannon’s entropy has nothing to do with the spatial behaviour of the nesting sites; if we rearrange the pixels in a different spatial configuration and compute Shannon’s entropy again, the result is the same, as it only depends on $p(x_{1}^{\left(discr\right)})$ and $p(x_{0}^{\left(discr\right)})$.

The computation of Shannon’s entropy for the two data types shows that this measure offers several possibilities over the same data, depending on the objectives of the study. As a consequence, when drawing conclusions, one should be careful about the variable under study and the correct interpretation of the results. In this example, if the aim is to detect the nesting habits of gorillas over the area, paying attention to the spatial structure or the effect of certain covariates on these habits, the usefulness of this index alone appears to be very weak.

### 2.3. Comments on the Effect of the Grid Size

We run an empirical sensitivity study of Shannon’s entropy computed on the variable $X^{\left(discr\right)}$ for the grid data. We consider different grid resolutions, ranging from 10×10 to 500×500 pixels. The results are reported in Figure 4.

In the left panel, the variable *number of nests* (*N. nests*) on the *y*-axis counts the number of pixels with at least one nest for each grid resolution. By increasing the resolution (*x*-axis), the variable *N. nests* appears to converge to the black horizontal line corresponding to the actual number of nests n=647. This is confirmed by a well-known result in point process studies, i.e., that the discretised process converges to the true one as the grid resolution increases [18]. In general, the implication is that the finer the grid, the more reliable the results, which is true in model-based approaches [19]. Unfortunately, the consequences of the descriptive entropy are not that straightforward. The middle panel shows that the variable *frequency of nests* (*Fr. nests*), that is, the proportion of pixels with at least one nest over the total number of pixels, tends to zero as the resolution increases. Indeed, as the grid becomes finer, the number of pixels with a nest slowly increases, but the number of pixels with no nests explodes, causing the relative frequencies of nests to become negligible. As a consequence, Shannon’s entropy for the grid data, shown in the right panel in relative terms, also degenerates to zero. A zero-valued entropy should be interpreted as a lack of diversity, which in this case, means that one is sure *not to observe* gorilla nests over the area. However, this value does not correspond to a real absence of gorilla nests and must be avoided. As a suggestion, it is advisable to use the original point data whenever possible. If not, a trade-off between the accuracy of the resolution and the reliability of the results is needed. We propose choosing the grid resolution that corresponds to the resolution of the available covariates. This choice is marked by the vertical red lines in all plots; it may represent a good compromise, as the relative frequency does not degenerate to zero, and the available covariate information can be matched to pixels and exploited in the following spatial measures.

When Shannon’s entropy is computed over the gorilla point data, the scale-dependence issue is not relevant, as points are assumed to be a-dimensional and represent the original data resolution, while the observation window area is exogenously fixed by the administrative boundaries of the gorilla sanctuary area. Therefore, there is no arbitrariness in the computation of Shannon’s entropy on the gorilla point data. This is further discussed in Section 6.

## 3. Partition-Based Spatial Entropy

In order to include information about the spatial configuration of the nesting sites, we need to modify Shannon’s entropy using one of the proposals available in the literature.

The first index involving space is Batty’s entropy [4,20], recently discussed in [21]. It starts by considering a single phenomenon of interest *F*, in our example the occurrence of a nesting site, and partitioning the observation window into *G* sub-areas of interest. Typically, sub-areas are meaningful for the problem at hand, e.g., administrative boundaries or patches defined by the values of a spatial covariate. It is possible to build a random partition in any number of sub-areas; however, this is not desirable for interpretation, as all conclusions are affected by the choice of the partition, which should be well grounded. The idea is to evaluate the distribution of such a phenomenon over the sub-areas, taking into account the size of the sub-area $T_{g}$, where $\sum_{g}Tg=T$. The *intensity* of the phenomenon over a sub-area is defined as $\lambda_{g}=p_{g}/T_{g}$, where $p_{g}$ is the probability of observation over area *g*, with $\sum_{g}p_{g}=1$. Batty’s entropy can be computed as
(2)$$H_{B}\left(F\right)=\sum_{g=1}^{G}p_{g}\log\frac{1}{\lambda_{g}}$$
and is interpreted as follows: if the phenomenon is equally intense over the window, i.e., if the intensities $\lambda_{g}$ for g=1,…,G are similar, Batty’s entropy is high, indicating maximum spatial diversity; if the phenomenon is concentrated over one or a few sub-areas, Batty’s entropy decreases. The index is in the range of $\left[\log T_{g*},\log T\right]$, where g* indicates the smallest sub-area. Note that computational issues may arise if the size of the sub-area is smaller than 1 (negative logarithms). This can be addressed by rescaling the observation area in order to have $T_{g}>1$ for all *g*. When the original area is multiplied by a factor of c>1 so that each rescaled sub-area has a new size equal to $T_{g}\times c$, Batty’s entropy can be derived by computing the entropy on the rescaled area and then subtracting logc from the resulting entropy value. After this transformation, interpretation proceeds in the same way, referring to the size of the original area. Such rescaling is now automatically run in the batty function of the SpatEntropy package.

A modification of Batty’s entropy introduced the idea of neighbourhood and distance between sub-areas (see [21] for a discussion). It was initially proposed by Karlström and Ceccato [5] and can also be called Batty’s LISA (Local Indices of Spatial Association) entropy [22]. This entropy discards the area size and introduces weights ${\tilde{p}}_{g}$ in the computation, i.e., the probabilities of the neighbouring sub-areas, where the neighbourhood extent must be fixed by the researcher. In this way, a sort of *smoothing* is obtained over the observation area
(3)$$H_{\mathrm{LISA}}\left(F\right)=\sum_{g=1}^{G}p_{g}\log\frac{1}{{\tilde{p}}_{g}}.$$
This entropy is in the range of [0,logG], regardless of the area partition. Note that since the size of the sub-area is discarded, no computational issues arise in the LISA version.

### 3.1. Computations on the Motivating Example

Batty’s entropy and its LISA variant are computed on the unmarked pattern, as they are unable to account for different data categories. Separate measures conditional on each category, i.e., ‘major’ and ‘minor’ group sizes, can be computed, but no joint result is possible. In the present study, we focus on the overall pattern. The entropies can be run in their basic version in SpatEntropy with:


> HB = batty(ungorillas)


 > HL = battyLISA(ungorillas, neigh = 1)
If not specified, a random partition in G=10 sub-areas is automatically performed by the function: 10 points are randomly generated over the polygonal window and are used as the centroids of the sub-areas; then, a Voronoi tessellation [23] is performed around those points. Random partitions are mainly intended for practice using the function in R or for situations with no alternatives, and disseminating the results is not recommended. Nevertheless, note that the results still make some sense: if a dataset comprises randomly scattered points over the area, any random partition should lead to the same conclusion of high diversity.

In real studies, a more appropriate choice is to identify a partition into sub-areas with a meaning for the phenomenon under study. For the gorilla nesting site data, it makes sense to use the covariates as the bases of a partition, as shown in Figure 5. The categories of the covariate *vegetation* can be employed: the window can be divided into four sub-areas (‘Disturbed’, ‘Grassland’, ‘Primary’ and ‘Secondary’) to evaluate whether the intensity of the nesting site construction varies across different vegetation categories, i.e., if the type of vegetation affects the frequency of the nesting sites. Looking at the second panel in Figure 5, it seems that gorillas have a strong preference for building nesting sites across the vegetation type ‘Primary’. Now, let us build Batty’s entropy by typing


> HBveg = batty(ungorillas, partition = veget)


 > HLveg = battyLISA(ungorillas, partition = veget, neigh = 1)
Let us now do the same with the covariates *elevation* and *water*; being continuous variables, they have to be divided into classes in order to create tiles for the area partition; e.g., classes may be defined by the distribution quartiles:


> HBelev = batty(ungorillas, partition = cut(elev,


      breaks = quantile(elev$v, na.rm = T)))
The computation of Batty’s LISA entropy for *elevation* and both indices for *water* proceed analogously (the code can be found in the Appendix A). For the LISA entropies, we fix neigh = 1 to include the probabilities of the nearest neighbour of each area; the function default neighbourhood system includes the nearest four sub-areas, measured in terms of Euclidean distances between centroids.

The computational times for both indices across all options are negligible (reported in Table 6) and the results are delivered in less than 1 s. The function output shows the absolute and relative values of the entropy and the object areas, with details about the quantities of interest for the given area partition: the absolute and relative frequencies (estimate for $p_{g}$), the area size $T_{g}$, and the intensity $\lambda_{g}$. The function has the new argument rescale, by default, set to TRUE, which automatically detects any $T_{g}<1$, performs a rescaling of the sizes of the areas, computes the entropy on the rescaled areas, and then transforms it back to refer to the original data. A plot similar to the panels shown in Figure 5 is also part of the output and is a novelty of the latest package version to help deliver results and assist in interpretation (when re-running the code, allow for differences in the area partition of the left panel since it is randomly generated). The results for the different partition options are reported in Table 2. For the random partition option, we also perform an empirical sensitivity study with regard to the number of sub-areas and the randomness of the generation. We choose several values for G=2,4,6,10,15,20,30,40,50,60,80,100, and for each *G*, we produce 1000 random area partitions. Then, we compute Batty’s and Batty’s LISA entropies for all generations. Figure 6 shows the mean value for each entropy and each *G*, as well as the empirical 95% confidence intervals over the simulations.

### 3.2. Interpretation of Results

The value for Batty’s entropy measures the level of concentration (low values) or dispersion (high values) of the gorilla nesting sites across sub-areas. Its LISA version measures the same, with the simultaneous consideration of the nesting behaviour in the neighbouring sub-areas.

Table 2 summarises the entropy values for Batty’s and Batty LISA entropies in relative terms for all partition options so that they can be compared across different values of *G* and, if desired, to the other entropy measures. Table 3 focuses on the meaningful area partitions based on the covariates and offers details about the quantities for each sub-area. Figure 6 analyses the entropy values across different random partitions. If we focus on the original version of Batty’s entropy, we can see that its value over the three covariate-based partitions is pretty stable and very high at 94% or more of the maximum entropy. A similar (mean) value is reported for the random partition in 10 sub-areas in Table 2. If we investigate the results of the simulation study reported in Figure 6, we can see that the entropy is hardly sensitive to both the number *G* of sub-areas and the random generation of the tessellation. Indeed, the mean values for Batty’s entropy start at 99% of the maximum for G=2 and slowly decrease to 93% for G=100. The segments representing the empirical confidence intervals at a 95% level are hardly visible, marking very little variability of the index. Its robustness with respect to the number and randomness of the area partitions may be considered an advantage of the measure. The very high values in relative terms indicate that quite a large diversity of the nesting sites across sub-areas is observed, i.e., that gorillas tend to build nests over all sub-areas without a preference. The similarity in the values for the random partitions and the covariate-based partitions suggests that the covariates play no role in the distribution of the gorilla nesting sites. This is in contrast to the visual impression, especially the vegetation panel in Figure 5, where it is evident that gorillas have a preference for the ’Primary’ vegetation type. This probably happens because within the primary vegetation area, there are actually many small spots where the vegetation is of a different type but nests still occur. Therefore, according to the vegetation partition, nesting sites are spread across the different categories, whereas actually, it is likely that the tendency of gorillas is to build nests across or *close to* primary vegetation areas. Batty’s entropy cannot account for a preference to be *close to* one sub-area, but we can try to improve the results with its LISA variant.

When a neighbourhood system among the sub-areas is considered, i.e., when the nesting habits are analysed with the consideration of the habit in the closest sub-area, the entropy decreases for all partition options, except for the one based on the covariate *water*. In other words, when we take the neighbouring nesting behaviours into account, a tendency appears to concentrate nests into one or a few close sub-areas. As for the random partition, which is not meaningful for the dissemination of the results, it is nevertheless interesting to see in Figure 6 that the mean values are quite stable and slightly increase when *G* increases, with the opposite behaviour with respect to Batty’s original entropy. Unfortunately, the addition of a neighbourhood system largely increases the variability of the results, especially for small *G*s, so the conclusions are linked to higher uncertainty and are less reliable. It is particularly interesting to focus on the partition based on the vegetation covariate. The simultaneous consideration of the vegetation and probabilities of one nearest sub-area heavily decreases the entropy value to $H_{\mathrm{LISA}\;\mathrm{rel}}\left(F\right)=0.48$ (Table 2). Such a low entropy value suggests that gorillas tend to concentrate across or close to one of the sub-areas, and we can check, by looking at Table 3, that the higher values concern the sub-area ’Primary’. Therefore, knowing the vegetation type helps in predicting where nesting sites will be constructed. A weaker neighbourhood effect can be seen for the partition based on the covariate elevation, which decreases the entropy from 98% to 85% of the maximum (Table 2), with higher probabilities linked to higher elevation levels (Table 3). The water distance appears to have no effect on the distribution of nests, as can be seen in the relative entropy being 95% of the maximum, as shown in Table 2, and the evenly spread probabilities, as shown in Table 3.

In conclusion, we can say that following Batty’s approach, gorillas show a preference for building nests close to primary vegetation areas and at higher altitude levels, while apparently, they show no interest in building nests following the water courses (which are probably sufficiently spread across the area). As for the performance of the indices, we can see that Batty’s entropy is robust but not easily interpretable, whereas the LISA version allows for more sensible conclusions but is linked to higher uncertainty. Increasing the neighbourhood extent increases the LISA entropy for all partition options since the number of neighbours approaches the total number of four sub-areas, and the spatial smoothing becomes too rough for a useful interpretation.

## 4. Distance-Based Entropy Measures

A different approach to the inclusion of space in entropy measures is based on a consideration of the distances between occurrences of the categories, as well as an evaluation of the heterogeneity of couples/triples/sets of occurrences (named co-occurrences) at the chosen distance. A comparative presentation of the approach can be found in [3,8]. Starting with a categorical variable *X* with *I* categories, the variable denoting the types of co-occurrences is usually named *Z*. In the case of couples, the number of categories of *Z* is $I^{2}$, as all possible ordered matchings between categories within *X* are considered. Further sets of m>2 occurrences return a variable with $I^{m}$ categories and are not considered in this work.

An index based on co-occurrences was first proposed by O’Neill and coauthors [6]. It considers grid data and contiguous couples, i.e., all possible pixel couples within the grid that share a border. The relative frequency of each couple of categories $p^{\left(c\right)}\left(z_{r}\right)$ (where (c) stands for contiguous) is computed for all types $r=1,\cdots ,I^{2}$. Then, O’Neill’s entropy is
(4)$$H_{O}\left(Z\right)=\sum_{r=1}^{I^{2}}p^{\left(c\right)}\left(z_{r}\right)\log\frac{1}{p^{\left(c\right)}\left(z_{r}\right)}$$
and is in the range of $\left[0,\log I^{2}\right]$. The maximum is reached when all possible couples of categories within *X* are equally represented, linked to the maximum surprise in observing a new couple. The main advantages of this measure with respect to Batty’s partition-based approach are that the categories of the original variable are considered in the computations and the interactions among different categories are studied via the heterogeneity of the couples.

O’Neill’s seminal proposal has generated other contiguity-based indices, such as the relative contagion index [24], which is $1-H_{Orel}\left(Z\right)$, and Parresol and Edwards’ entropy [25], i.e., $-H_{O}\left(Z\right)$.

A stimulating extension of O’Neill’s entropy is due to Leibovici and coauthors [7,26]. It substitutes the notion of contiguity with that of distance by considering couples (or further sets of co-occurrences) taking place *within a chosen distance d*, which can also apply to point data. Leibovici’s entropy is
(5)$$H_{L}\left(Z\right)=\sum_{r=1}^{I^{2}}p^{\left(d\right)}\left(z_{r}\right)\log\frac{1}{p^{\left(d\right)}\left(z_{r}\right)}$$
and can potentially be computed for any distance within the observation window, according to the researcher’s choice. The substitution of contiguity with distance also allows for the computation of this measure on point data. The interpretation is similar to that of O’Neill’s entropy, except for the notion of distance, and the range is the same.

It has been proven [3] that O’Neill’s and Leibovici’s entropies are special cases of a quantity known in information theory as residual entropy [27]. Indeed, they must be interpreted as measuring the residual amount of information brought by the observations once the spatial information has been accounted for. In other words, they measure the level of diversity in the data that is not due to the spatial structure. The distance-based entropies are low when couples tend to be of the same type, which may happen because there is mutual attraction or repulsion between the different categories and/or within each category or because one of the categories is largely predominant so that couple heterogeneity is not possible. A high value means that couples are very heterogeneous, which implies that all categories are quite evenly present and there is no recognisable behaviour in the interaction between points. This suggests an absence of spatial structure. If one prefers to focus on the amount of spatial information rather than the residual diversity, contagion indices must be used for the delivery of results: the relative contagion index by construction must be interpreted in the opposite way.

### 4.1. Computations on the Motivating Example

To compute O’Neill’s entropy and its modified versions on the gorilla nesting site data, we need to work on discrete data $X^{\left(discr\right)}$, i.e., the pixel grid with the values ‘nest’ and ‘no nest’. Therefore, the variable $Z^{\left(discr\right)}$ has $I^{2}=4$ possible categories: ‘nest-nest’ (1-1), ‘nest-no nest’ (1-0), ‘no nest-nest’ (0-1), and ‘no nest-no nest’ (0-0). The entropy indices can be obtained by typing:


> oneill(discrgorillas$v)



> contagion(discrgorillas$v)


 > parredw(discrgorillas$v)
Thanks to efficient computations based on combinatorics, these functions require very little time and return results almost instantly (see Table 6). The output consists of the entropy value in absolute and relative terms in a table containing the couples’ relative frequencies and a plot. All values are reported in Table 4, while the plot shows the discrete data, as in the right panel in Figure 2.

Leibovici’s entropy allows for the introduction of the notion of distances between pixels or points. Before computing this entropy, it might be interesting to explore the distribution of distances between nesting sites, which are meaningful for the evaluation of the interaction within and between categories. The most used distances are the *nearest neighbour distance*, i.e., the distance between each point and its closest point, and the *pairwise distance*, i.e., the distances between all possible pairs of points [23]. They can be calculated via the spatstat functions nndist(ungorillas) and pairdist(ungorillas). A box plot of the distribution of these distances is shown in the left panel of Figure 8. The distance for computing Leibovici’s index may be chosen based on the aforementioned distance distributions.

Assuming one is working based on the *pairwise distance* using the variable $X^{\left(discr\right)}$ with the binary values ‘nest’ and ‘no nest’ for each pixel, Leibovici’s entropy can be obtained by typing


> leibovici(discrgorillas$v,



    cell.size = c(discrgorillas$xstep, discrgorillas$ystep),


      ccdist = median(c(pairdist(ungorillas))))
This function requires about 5 min (see Table 6) because of the computational burden of all Euclidean distances. With this large discrete dataset, a total of $n^{\left(d\right)}\times \left(n^{\left(d\right)}-1\right)=$ 21,062 × 21,061 distances must be calculated. For checking the progress of the function, the option verbose = T can be added to the previous command lines. The function output is again composed of absolute and relative entropies and a probability table for investigation of the results, as shown in the right-hand side of Table 5. Moreover, a plot is produced, which is reported in the right panel of Figure 7. The latter represents the data with an example of the extent of the distance considered for building couples. The red star represents a random point, and the radius of the red circle represents the spatial extent for counting the co-occurrences. This may help in guiding scientists towards a suitable choice.

An alternative and much faster option consists of computing Leibovici’s index on the data points by considering the categories of the two sizes of the variable *group*, i.e., the variable $X^{\left(\mathrm{cont}\right)}$. In this case, the function is

 > leibovici(gorillasgroup, ccdist = 500)(where the distance is in metres, the same unit as the observation area), which returns results in less than 1 s (see Table 6), as it only needs to compute n×(n−1)=647×646 pairwise distances. The function output is analogous, with details provided on the left-hand side of Table 5, and the plot can be found in the left panel of Figure 7 with the same features as in the discrete case.

### 4.2. Interpretation of the Results

O’Neill’s entropy, the contagion index, and Parresol and Edward’s entropy are closely related and based on the relative frequencies of the different possible couples of contiguous pixels. Table 4 reports the quantities and values of the indices.

The resulting value for O’Neill’s entropy over the discrete gorilla data was very low, with $H_{O\;\mathrm{rel}}\left(Z^{\left(discr\right)}\right)=0.17$. This tells us that contiguous couples tend to be homogeneous, i.e., that there is little diversity in the type of couples of adjacent pixels that appear across the area. As can be seen in Table 4, this is due to the large predominance of 0-valued (‘no nest’) pixels, which produced a large predominance of couples of type (0-0) (grey pixels in Figure 2). With such a minority of pixels with nests, we cannot reasonably conclude whether gorillas tend to build nests close to other nesting sites (couples 1-1) or in isolated areas (couples 1-0 and 0-1). The same reasoning underlies the values of the other two entropy indices. The contagion indices indicate that the strength of the contagion, i.e., the similarity between couples (as opposite of diversity), was equal to 83% of its theoretical maximum. As for Parresol and Edwards’ entropy, the value in relative terms was the same as O’Neill’s value and the interpretation also coincides. Therefore, despite the interesting theoretical aspects of the indices, such as the consideration of the variable categories and interpoint distances, their strength in delivering conclusions on the gorilla data does not appear convincing because they are heavily affected by the imbalance in the original distribution of $X^{\left(discr\right)}$.

Let us now focus on Leibovici’s entropy. The middle and right-side panels in Figure 8 show the variations in this entropy in relative terms by setting different distances. For discrete data (in the middle panel), distances were chosen between contiguity (i.e., the distance between two contiguous pixels’ centroids) and the maximum distance inside the polygonal window. For point data (in the right panel), the deciles of the pairwise distance distribution were considered.

The two panels returned different values, as they were computed on different datasets. For grid data, the heterogeneity of a couple of categories, ‘nest’ and ‘no nest’, was evaluated. The middle panel shows a very low relative entropy value (about 0.2). The interpretation follows what has already been said for O’Neill’s entropy, except that the more general concept of distance replaces that of contiguity: the large predominance of couples of type 0-0 affected the results across all distances, as seen in the detailed values in Table 5. Again, our comments cannot be very conclusive for the gorilla grid data with distance-based measures.

The interpretation improved when moving the original point data. The right panel shows Leibovici’s relative entropy for point data, evaluating the diversity of the group size with the ‘minor’ and ‘major’categories. The high relative entropy (approaching value 1) means that at the chosen distances, pairs of categories tended to be heterogeneous, i.e., groups may be close to other groups of the same or different size with no evidence for a preference, so that all couples were (nearly) equally represented and the diversity of the system was high with respect to the group size of the nests (see the values in Table 5). This suggests a random spatial structure of the group size, meaning that groups tended not to cluster or repel each other based on their size. In both panels, it can be seen that the relative index was quite stable, meaning that the conclusions were not affected by the chosen distance. Should this be the case, it would be important to choose the distance with care or explore and show results across different distances, as in the present work. The easy implementation of such entropy indices in the SpatEntropy package allows for the exploration of many possibilities in a reasonable time (see Table 6) and with minimum effort.

### 4.3. Comments on the Effect of the Grid Size

In order to raise awareness about issues related to the choice of grid resolution, we empirically address the scale-dependence problem for spatial measures using grid data. We specifically examine O’Neill’s entropy, the relative contagion index, Parresol and Edward’s entropy, and Leibovici’s entropy when applied to grid data. The same sensitivity analysis employed in Section 2 is used here, with grid resolutions ranging from 10×10 to 500×500 pixels. Therefore, one may refer to the number and frequency of pixels displayed in Figure 4. In Figure 9, we report the results for O’Neill’s relative entropy (also known as the relative Parresol and Edwards’ entropy), the relative contagion index, and Leibovici’s entropy with the second and the second-last distances chosen for the results in Figure 10, i.e., d=656 and d=4946 metres. The vertical red lines indicate the grid resolution chosen for the present study, i.e., the same resolution as the covariates.

The figure shows that all entropies exhibited the same behaviour, with the only difference being the contagion index due to the fact that this index is computed as the complement to 1 of O’Neill’s relative entropy. The interpretation is the same for all measures: at very high grid resolutions, all entropies tended to zero (or 1) due to the fact that the relative frequency of pixels with nests decreased significantly. The behaviour of the distance-based entropies was very similar to that of Shannon’s entropy (in Figure 4), as already shown in [3]. The same holds for the decomposable entropy measure in the following section. This study highlights that results are extremely sensitive to grid resolution. If possible, the original point data should be considered, as well as the corresponding entropies defined for point locations, i.e., on variable $X^{\left(cont\right)}$. This is possible for Leibovici’s entropy, but one must remember that in this study, the two entropies were computed on different variables, requiring separate conclusions. Some final comments on this point are given in Section 6.

## 5. A Decomposable Entropy Measure

Recently, a new approach to distance-based entropy has been introduced [8] and widely used in several applications [3,16,21]. It is based on the differences in building the variable *Z* and on the decomposition of Shannon’s entropy into spatial and non-spatial components. The variable *Z* is built with unordered pairs, rather than ordered couples, of categories within *X*. This might be more sensible, provided that occurrences do not have a direction in space; speeds up computations, as the number of pairs is reduced and equal to I+12; and returns a Shannon’s entropy of *Z*, H(Z) with the same value, regardless of the spatial arrangement of observations. A discussion about these aspects can be found in [3].

The non-spatial feature of H(Z) allows for the exploitation of a result in information theory [27]. Shannon’s entropy of a variable, in our case *Z*, is known to be decomposable into the entropy due to its relationship with another variable and a residual entropy term. If the second variable is defined as *W*, representing space, with categories $w_{1},\cdots ,w_{K}$ representing classes for all possible distances within the observation window, then the two terms of the decomposition may be interpreted as *spatial mutual information*, denoted by MI(Z,W), i.e., the diversity due to the spatial structure of the observations, and the *spatial residual entropy*, $H{\left(Z\right)}_{W}$, measuring all other sources of heterogeneity.
(6)$$\begin{array}{cc} H(Z) & =MI\left(Z,W\right)+H{\left(Z\right)}_{W}\\ \; & =\sum_{k=1}^{K}p\left(w_{k}\right)\left[PI\left(Z|w_{k}\right)+H\left(Z|w_{k}\right)\right]. \end{array}$$
The terms $p(w_{k})$ are the probabilities of each distance class, computed as the relative frequencies of the number of pairs occurring within the distance range $w_{k}$. $PI(Z|w_{k})$ and $H(Z|w_{k})$ are local or partial terms, expressing entropy related to space and residual entropy within each distance range $w_{k}$, respectively. Therefore, all distances are simultaneously considered in a decomposable measure, offering a global understanding while investigating the role of the spatial structure in predicting outcomes within chosen distance ranges, with the desired level of detail (chosen by fixing the number and extent of the distance classes).

The main advantage of the set of measures contained in (Equation 6) is its ability to adapt to different case studies, relying on the possibility of exploring different options, aspects of the same phenomenon, and exogenous choices, which may influence the results. In the recent literature, this approach is also known as *decomposable entropy* (the main reference can be found in [3]).

### 5.1. Computations on the Motivating Example

As for Leibovici’s entropy, the decomposable entropy function can be computed on both grid and point data. In the SpatEntropy package, this function is named altieri, after the family name of the authors who proposed it. The command lines for the two data types are as follows:


> altieri(discrgorillas$v,



      cell.size = c(discrgorillas$xstep, discrgorillas$ystep))


 > altieri(gorillasgroup)
For discrete data, the function requires about 10 min on such a large dataset for the same reasons as Leibovici’s entropy, which is linked to the computation of a large number of pairwise distances and the construction of the distance classes. For point data, the results are delivered in less than 1 s (see Table 6). A default partition for the distance classes is proposed in the function: for grid data, the first distance class covers the 4 nearest pixels, the second covers the 12 nearest pixels, and the third is the residual class for all the remaining distances. This follows common spatial statistics outcomes [28]. For point data, the default breaks are based on the deciles of the distribution of the nearest neighbour distances. These distance breaks can be modified by specifying the argument distbreak in the function.

This function’s output is not reported here, as the produced plot is sufficient for interpretation purposes. First, the global values for Shannon’s entropy of *Z*, the mutual information, and the residual entropy are given, and then the absolute and relative values of the partial mutual information and residual entropy terms are provided for the chosen classes. They are followed by information about the distance classes, such as the number of pairs for each class and their relative weights in the computation. Tables are also provided, showing the absolute and relative frequencies of all pairs for each distance class. We refer the reader to [16] for technical details on the output. The newly released package version automatically produces a bar plot, which shows the contribution of space in relative terms (grey area) for each distance range, thereby significantly helping in the interpretation. This bar plot is shown in Figure 10.

### 5.2. Interpretation of Results

Let us look at the bar plot for decomposable entropy in relative terms on the gorilla data in Figure 10. The left panel refers to grid data, that is, pixels with the same resolution as the spatial covariates, characterised by the presence (‘nest’) or absence (‘no nest’) of a nest. The right panel reports the results on point data, that is, locations of nests separated by the size (‘major’ or ‘minor’) of the gorilla group.

The main advantage of using this approach to spatial entropy is the ability to explore several definitions of distances among co-occurrences within a single analysis. In this example, we consider the three distance classes defined by the default options. Decomposable entropy allows disentangling the role of space at each distance (local partial information, grey area in the bars) from the entropy due to other features (partial residual entropy, white area in the bars). As a result, Figure 10 helps in visually understanding the behaviour of data very quickly: the larger the grey area, the more the spatial structure helps in understanding the diversity of data within a specific distance range, thereby reducing the residual entropy. In particular, it can be seen that some role of space can be observed for discrete data, i.e., pairs of pixels tend to be of the same type (mainly ‘no nest’-‘no nest’) across the whole observation window. In other words, this indicates that gorilla nesting habits have a non-negligible dependence on spatial structure; however, such interpretation suffers from an imbalance in the distribution of pixels, as already discussed in Section 4. In addition to the previous distance-based measures, we can see that spatial information appears to be consistent across the three distance classes depicted in the picture. Conversely, for point data, space is confirmed not to play any relevant role in the heterogeneity of pairs characterised by locations of ‘minor’ or ‘major’ sizes of gorilla groups across all distances. As a consequence, the residual entropies are set to very high values, suggesting the randomness of gorillas’ nesting habits across the region based on their group size. Similar to Leibovici’s entropy, the results are more conclusive and easily interpretable for point data.

### 5.3. Comments on the Effect of the Grid Size

A sensitivity analysis was performed on the decomposable entropy, analogously to that in Section 2 and Section 4. As already explained, all entropy measures based on *Z* and applied to grid data were affected by the grid resolution in the same way. The decomposable entropy also tended to zero as the relative frequency of the pixels with nests decreased. Therefore, the same conclusions can be drawn as in Section 4, and the study is not reported here. Some final comments are given in Section 6.

## 6. Discussion

This section contains some points for discussion, as well as suggestions for further work before we give our concluding comments in Section 7.

### 6.1. About the SpatEntropy Package

The SpatEntropy package allows for the implementation of all entropy measures using intuitive and simple command lines in a short time. A summary of the computational times for all measures is reported in Table 6.

The reported features refer to the application to the gorilla data, which can be considered a large dataset. Indeed, the enclosing rectangle is a very fine grid of nearly 27,000 pixels and, consequently, the number of pairwise distances across co-occurrences to compute becomes huge. Despite this, the computational time required by the package remains reasonable, thereby avoiding the undesirable need to aggregate data, thus losing information and reducing image quality. The yielded results are easily interpretable and intuitive, thanks to the novel production of output plots. These encourage the diffusion of the package and the dissemination of the results while supporting interpretations for decision-making processes.

### 6.2. A Summary of Spatial Entropy Measures

Throughout this paper, we showed that each of the considered entropy measures may be useful according to the context of the study at hand, each having various advantages and drawbacks. We utilised an application to a real dataset on gorilla nesting sites to highlight these features. In particular, the results on data from the original point pattern (focusing on the *group* mark, with outcomes ‘major’ or ‘minor’) and from a grid (obtained by discretising the observation area into pixels with at least one ‘nest’ or ‘no nest’) were provided to showcase the capabilities of entropy measures with both kinds of data. To assist researchers in selecting from the available indices, Table 7 reports a summary of their main characteristics and usage.

The decision to work with grids of pixels rather than point patterns depends on data availability and/or the purposes of the study. In the gorilla dataset, the two types of data have different research questions: the point data aim to investigate diversity in the spatial locations of nests with respect to the size of the gorilla group, whereas in the grid, the distribution of the presence of nests over the area is studied. Consequently, space can be included in the entropy measure in several ways. If the goal of the research is to measure diversity in the intensity of the phenomenon (the nests, in our motivating example) across specific and well-defined sub-areas of interest, then Batty’s and Batty’s LISA entropies are appropriate measures, controlling for sub-area dimensions or neighbourhood structures, respectively. In the gorilla data example, Batty’s LISA entropy seems to be more informative, i.e., the role of the neighbourhood appears to be relevant in predicting the nesting sub-area, thus yielding lower entropy values compared to Batty’s and Shannon’s values. If the finest location of the phenomenon, in terms of a point or pixel in a grid, is to be considered, then suitable entropy measures would be those based on the concept of co-occurrence. Under this framework, the most complete approach is decomposable entropy, as it enables disentangling the role of space and the residual part from other components, the concept of neighbourhood is generalised to a variable that expresses distance classes across the entire possible range, and local entropies at each distance class can be investigated to understand different aspects of the phenomenon. The results obtained from the gorilla data show that the role of space can be detected on the grid across all distance classes, i.e., the diversity in couples of ‘nest’/‘no nest’ pixels decreases when the neighbourhood is accounted for. Conversely, when point data on the nesting location by ‘major’ or ‘minor’ sizes of gorilla groups are considered, the spatial configuration is negligible across all distances, i.e., the entropy across the couples of ‘major’/‘minor’ groups mainly depends on features and choices different from the spatial position over the area (falling into the residual entropy component). Leibovici’s and O’Neills’s entropies are useful special cases of decomposable entropy, which can be conveniently delivered for simplicity of interpretation, especially when the researcher knows that a specific distance is particularly meaningful for the phenomenon under study.

### 6.3. Choice of Spatial Parameters

Batty’s and Batty’s LISA measures are affected by the choice of sub-areas and, for the LISA variant, the choice of neighbourhood, and it may be hard to decide which is the best option. Therefore, we recommend relying on a meaningful classification according to the knowledge of the phenomenon, such as administrative boundaries or covariate values, and/or checking and comparing the different options, which is very feasible thanks to the small computational burden in the SpatEntropy package.

When grid data were considered, we showed that the resolution affected the results of all heterogeneity measures. As reported in Figure 4 and Figure 9, entropies degenerated to zero as the resolution increased. In many studies, a grid is provided as the original data resolution. In these cases, identifying the contribution of this restricted definition of space represents a relevant research question. When different choices for space can be adopted, considering the highest available resolution is, in general, recommended, provided that the relative frequencies of the categories do not become negligible. Indeed, in the application to gorilla nesting sites, we showed that the chosen grid resolution represents a good trade-off, as the entropy values remained considerable. When spatial covariates are available, as in our motivating example, we recommend choosing the same resolution as the one provided by the covariates, as this allows for exploiting and incorporating the additional information into the analysis.

In the case of point data, generally, the observation window is fixed, as in the application to gorilla nesting sites. In these situations, O’Neill’s, contagion, and Parresol and Edwards’ entropies cannot be computed, and Leibovici’s entropy and the decomposable entropy are recommended for evaluating the interactions within and between data categories.

## 7. Conclusions

In this paper, some leading measures of spatial entropy have been outlined and applied to a dataset about gorilla nesting sites in Cameroon, based on the newly released version of the R package SpatEntropy. The entropy indices range from the simplest non-spatial Shannon’s entropy to the flexible framework of decomposable entropies [3,8], which is the most complete and properly grounded from a theoretical point of view.

We have conducted a detailed analysis of a stimulating and computationally challenging dataset regarding the nesting habits of gorillas in a protected sanctuary area in Cameroon, considering group size and some environmental covariates. The main conclusions are drawn from the analysis of the point data. A study based on Batty’s LISA entropy shows that the construction of the nesting sites is heavily influenced by the covariate vegetation, with a preference for building nests over or very close to the vegetation type ‘Primary’. A weaker effect of the soil elevation is also observed, with high altitudes appearing to be favoured by gorillas. No relevant effect is associated with the distance to water courses, probably because they are evenly distributed over the observation area. When focusing on the interactions within and between the different group sizes, Leibovici’s entropy and the decomposable entropy allow us to conclude that gorilla groups do not tend to cluster or repel each other according to their group size. Minor and major group sizes are randomly scattered over the area for all the considered distances, with no recognisable interaction. Results have also been shown for the discretised data, but they are less clearly interpretable and affected by the arbitrary choice for the grid resolution.

The contributions of this paper are application- and software-oriented. Its main aim is to provide a practical guide to researchers, firstly, for computing spatial entropy measures from the simplest ones to the more computationally complex indices based on co-occurrences, and secondly, for supporting the choice of proper measures, suitable space parameters, and correct interpretations of the results. In this regard, we would like to remark that the challenging problem of diversity indices for spatial data benefits from stimulating recent works that consider theoretical aspects. For instance, the use of data relative frequencies as proper choices for estimating probabilities is discussed in [17], where a model-based approach is taken to include covariates and spatial dependence in the entropy measures. Other computational details regarding the indices illustrated in this work, such as the choice of preserving order within couples, the consequences of extending to larger sets of co-occurrences, or the choice of the distance breaks in the decomposable measure, are discussed in [3]. Further aspects linked to the dependence of the indices on the scale and resolution of the data can be found in [9,12,13,14]. Defining spatial entropy measures satisfying the scale-free property represents a crucial task, especially when arbitrary choices of the spatial scale or resolution of data can be adopted.

## Figures and Tables

**Figure 1 entropy-25-01634-f001:**
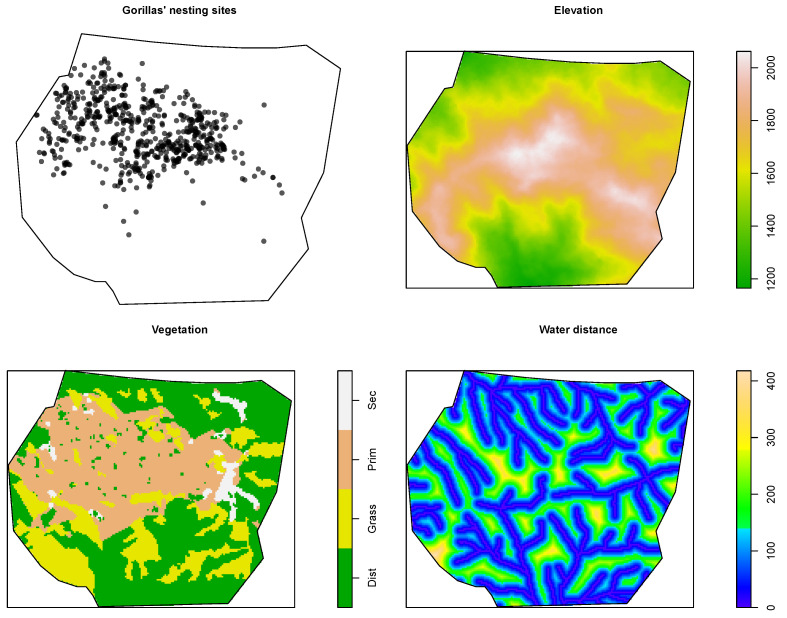
Gorilla nesting sites and environmental covariates.

**Figure 2 entropy-25-01634-f002:**
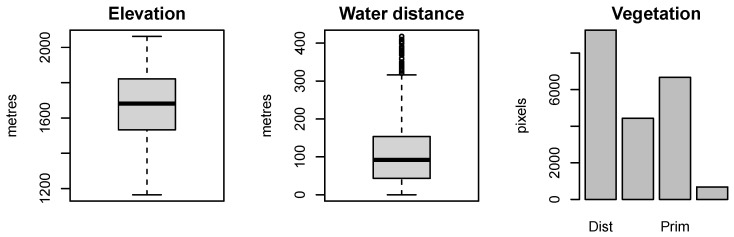
Covariate information.

**Figure 3 entropy-25-01634-f003:**
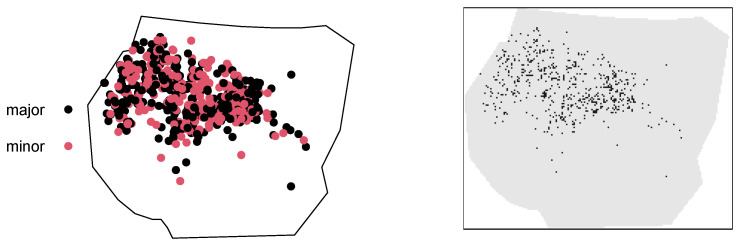
Point data with group marks (**left**) and unmarked grid data (**right**).

**Figure 4 entropy-25-01634-f004:**
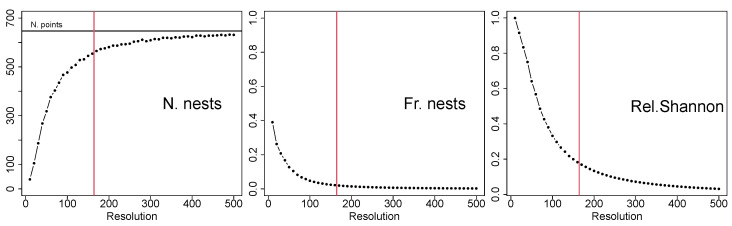
Effect of the grid resolution on the number of pixels with nests, their relative frequencies, and Shannon’s relative entropy. The vertical red lines mark the chosen resolution for the application.

**Figure 5 entropy-25-01634-f005:**
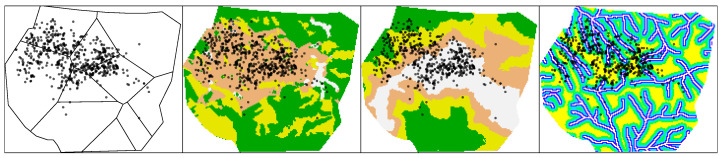
Partition options for Batty’s entropy: random partition, vegetation categories, elevation and water divided into distribution quartiles.

**Figure 6 entropy-25-01634-f006:**
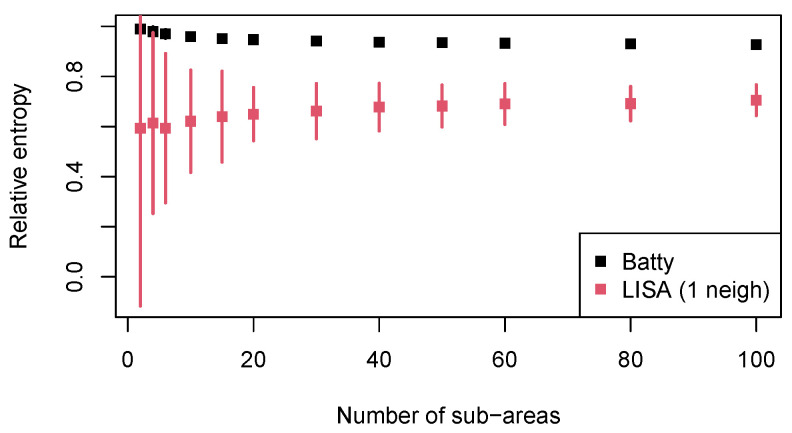
Relative Batty’s and Batty’s LISA entropies over 1000 random partitions for each number of sub-areas: mean values and empirical 95% confidence intervals.

**Figure 7 entropy-25-01634-f007:**
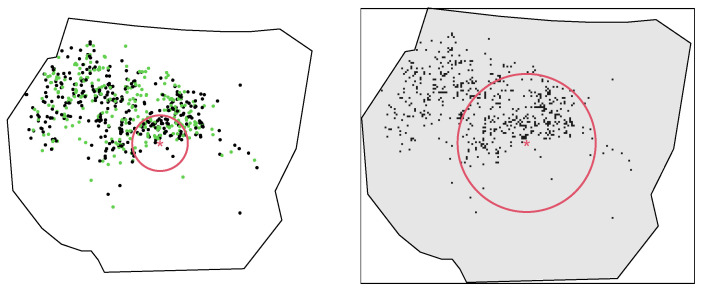
Example of distance for Leibovici’s entropy for point and grid data.

**Figure 8 entropy-25-01634-f008:**
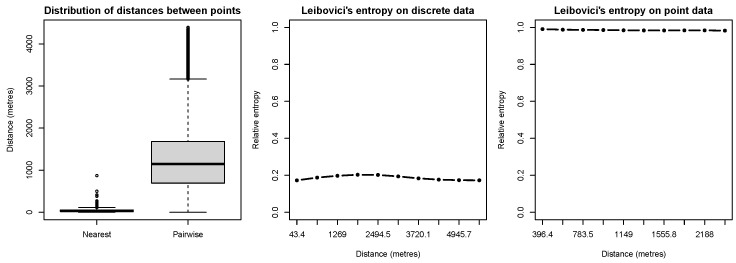
Relative Leibovici’s entropy for different distances.

**Figure 9 entropy-25-01634-f009:**
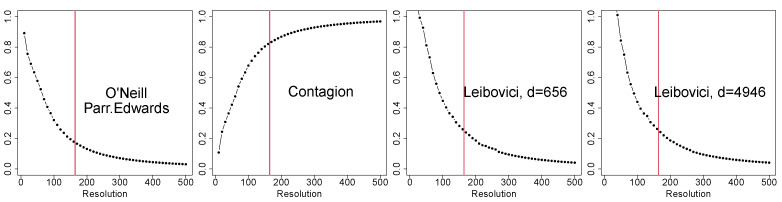
Effect of the grid resolution on the relative O’Neill’s entropy (equal to the relative Parresol and Edwards’ entropy), relative contagion index, and relative Leibovici’s entropy. The vertical red lines indicate the chosen resolution for the application.

**Figure 10 entropy-25-01634-f010:**
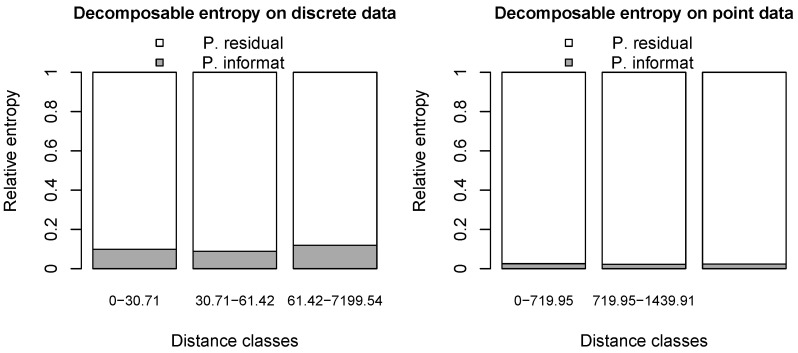
Decomposable entropy for gorilla data.

**Table 1 entropy-25-01634-t001:** Absolute and relative frequencies and relative Shannon’s entropy for point and grid data.

Point Data				Grid Data			
Category	Abs. Freq.	Prob.	Shannon	Category	Abs. Freq.	Prob.	Shannon
Major	350	0.54	0.99	0	20,513	0.97	0.17
Minor	297	0.46		1	549	0.03	

**Table 2 entropy-25-01634-t002:** Relative Batty’s and Batty’s LISA entropies with four partition options. * Mean value over 1000 simulations with G=10 (see Figure 6 for details).

Entropy	Random *	Vegetation	Elevation	Water
Batty	0.96	0.97	0.98	0.94
LISA	0.62	0.48	0.85	0.95

**Table 3 entropy-25-01634-t003:** Information about Batty’s entropy over the three covariate-based partitions: sub-area name, absolute and relative frequencies of nests, area size, and intensity.

Vegetation-Based Partition				
**Sub-area**	**Abs. freq.**	**Prob.**	**Size (km${}^{2}$)**	**Intensity**
Disturbed	89	0.14	8.72	1.58 × 10${}^{-8}$
Grassland	20	0.03	4.18	$7.39\times 10^{-9}$
Primary	517	0.80	6.29	$1.27\times 10^{-7}$
Secondary	21	0.03	6.43	$5.05\times 10^{-8}$
**Elevation-Based Partition**				
**Sub-area**	**Abs. freq.**	**Prob.**	**Size (km${}^{2}$)**	**Intensity**
Low	58	0.09	4.97	$1.80\times 10^{-8}$
Med-low	141	0.22	4.97	$4.39\times 10^{-8}$
Med-high	110	0.17	4.97	$3.42\times 10^{-8}$
High	338	0.52	4.97	$1.06\times 10^{-7}$
**Water Distance-Based Partition**				
**Sub-area**	**Abs. freq.**	**Prob.**	**Size (km${}^{2}$)**	**Intensity**
Adjacent	103	0.23	4.97	$4.63\times 10^{-8}$
Close	126	0.19	4.97	$3.82\times 10^{-8}$
Medium	155	0.24	4.97	$4.83\times 10^{-8}$
Far	223	0.34	4.97	$6.84\times 10^{-8}$

**Table 4 entropy-25-01634-t004:** Absolute and relative frequencies of contiguous couples, O’Neill’s, contagion, and Parresol and Edwards’ indices in relative terms.

Couple	Abs. Freq.	Prob.	O’Neill	Contagion	Parr.Edwards
0-0	39,775	0.95	0.17	0.83	0.17
0-1	919	0.02			
1-0	919	0.02			
1-1	179	0.01			

**Table 5 entropy-25-01634-t005:** Absolute and relative frequencies and relative Leibovici’s entropy for point data with d=500 and grid data with d=1149 (the median of the pairwise distances).

Point Data				Grid Data			
Couple	Abs. Freq.	Prob.	Leibovici	Couple	Abs. Freq.	Prob.	Leibovici
maj-maj	4334	0.28	0.99	0-0	18,262,284	0.95	0.18
maj-min	3780	0.26		0-1	501,725	0.02	
min-maj	3835	0.25		1-0	514,686	0.02	
min-min	3404	0.22		1-1	22,427	<0.01	

**Table 6 entropy-25-01634-t006:** Computational times for SpatEntropy functions: means and interquartile differences (IQDs) for the computational times over 1000 runs in seconds (on a 2019 Windows Surface Pro 6 with an i7-8650U processor).

Function	Data Size	Mean Time	IQD
shannon	647 points	0.0017	[0.0015; 0.0018]
shannon	26,969 pixels	0.0617	[0.0441; 0.0654]
batty	647 points	0.6349	[0.4317; 0.7807]
battyLISA	647 points	0.7536	[0.4673; 0.9606]
oneill	26,969 pixels	0.9012	[0.6816; 1.0784]
contagion	26,969 pixels	0.8612	[0.6460; 1.0490]
parredw	26,969 pixels	0.8013	[0.6493; 0.9770]
leibovici	26,969 pixels	252.1926	[221.8776; 274.3218]
leibovici	647 points	0.8142	[0.6518; 0.8803]
altieri	26,969 pixels	566.3254	[418.8298; 761.0572]
altieri	647 points	0.9029	[0.8826; 0.9142]

**Table 7 entropy-25-01634-t007:** Summary of main features and applicability of entropy measures. * The time may vary according to the data size—see Table 6 for details.

Entropy	Grid/Point	Categ.	Cov.	Space	Scale Dep.	Time *
Shannon	Both	Yes	No	Not included	Yes (grid)	<1 s
Batty	Both	No	Yes	Area partition	Yes (grid)	<1 s
Batty LISA	Both	No	Yes	Area partition	Yes (grid)	<1 s
O’Neill	Grid	Yes	No	Contiguous couples	Yes	≤1 s
Leibovici	Both	Yes	No	Dist-based sets	Yes (grid)	≤ 5 min
Decomposable	Both	Yes	No	Dist-based sets	Yes (grid)	≤ 10 min

## Data Availability

Data are free to download by installing the software R and then typing install.packages(spatstat); library(spatstat); data(gorillas). The R code for reproducing all results is reported in the present paper and available in the Appendix A.

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
