# Peer review of "Efficient Computation of Spatial Entropy Measures"

_entropy, 2023, doi:10.3390/e25121634_

Round 1

Reviewer 1 Report

Comments and Suggestions for Authors

General Comments

In this paper, the newly released version of the R package SpatEntropy was employed to make an entropy-based study on the spatial distribution of gorilla nesting sites in Cameroon. The main conclusion drawn in this study seems to be that distance-based entropies as proper measures of biodiversity. Due to the dependence of entropy values on the number of states and the uniformity of distribution, entropy has always been used as a measure of diversity and complexity. However, measuring spatial entropy involves a challenge, which is scale dependence of entropy values on regional sizes or spatial distance. Changing the scale of spatial measurement or the definition of regional space will result in a change in entropy values. Unfortunately, the author did not have any discussion on the scale dependence of spatial entropy.

This article mainly presents the application results of a technology, that is, the R package SpatEntropy. In terms of theory, methods, and empirical analysis, this article lacks advantages for the time being.

The subject addressed in this article is worthy of investigation. Organization of the manuscript is not appropriate. Due to the structural flaws of the paper, the paper is not acceptable for publication in its present form.  But it can be reconsidered after major revision. I suggest that the authors modify the manuscript carefully by referring to the following comments.

Specific suggestions

First, the Abstract should be improved. A good abstract contains four elements: background, methods, results, and conclusions. The abstract had better been revised according to the following structural arrangement: background and aim, methods, results or findings, conclusions or significance.

Second, the structure of the Discussion part of this paper is incomplete. The discussion in the paper is not in-depth enough. The section of Results represents the heart of a paper, and the section of Discussion is the paper’s nerve center. The section of Discussion in a paper is generally involved with 3 or 4 parts: (1) main points, which response to the questions put in introduction; (2) comments on related studies or problems; (3) shortcomings or deficiency in study method or process; (4) conclusions, which can be separated to make the final section.

Third, the scale dependence of spatial entropy was not discussed. In this paper, Figures 3, 4, and 6 suggest scale-dependence or scale-free property. Due to the scale dependence of spatial entropy, in many cases, spatial entropy should be converted into an information dimension according to the scaling law. [See for example, Chen Y, Equivalent relation between normalized spatial entropy and fractal dimension. Physica A, 2020, 553, 124627; Chen Y, Wang J, Feng J. Understanding the fractal dimensions of urban forms through spatial entropy. Entropy, 2017, 19, 600]. Michael Batty proposed the concept and measure of spatial entropy. However, 20 years later, he advocated the concept of fractal dimension for spatial analysis [See for example, Batty M, Longley PA, Fractal Cities. London: Academic Press, 1994]. In fact, in many cases, entropy is associated with fractal dimension. [See for example, Ryabko BYa (1986). Noise-free coding of combinatorial sources, Hausdorff dimension and Kolmogorov complexity. Problemy Peredachi Informatsii, 22: 16-26; Sparavigna AC, Entropies and fractal dimensions. Philica, 2016, hal-01377975]

Fourth, the conclusions should be made clearer in expressions and meanings. The important conclusions should be given three times in a paper: once in the Abstract, again in the Introduction, and again (in more detail probably) in the Discussion (if it contains a Conclusion paragraph) or Conclusions (if this part is separated as the final section).

Fifth, conclusions are confused with results in this paper. Conclusions are different from results. Generally speaking, results come from or are directly based on data analysis. In contrast, conclusions come from discussion and represent the climax of discussion. If the result is regarded as the heart of an academic paper, the discussion can be treated as the nerve center of the paper. Discussions form a bridge between results and conclusions.

Sixth, it ignores quite large sections of the spatial entropy literature, particularly developments in the spatial entropy research on scale-free distribution.

Comments on the Quality of English Language

The English language is acceptable.

Reviewer 2 Report

Comments and Suggestions for Authors

General
=======

The paper is essentially a user manual for an R-package computing
spatial entropy measures. It gives a structured overview on the
available features and shows with examples how they can be used. It
tells us what is computed precise enough to identify which of several
possibilities it is, but not precise enough to know all relevant
parameters such that results could be reproduced. I.e. e.g. does not
explain how areas are separated, or what would happen with non
rectangular observation windows. It however references orginal papers
like someone would find it in a software documentation. At several
instances the fast computing speed is mentioned and given in terms of
computing time without precisely specifying the computational time,
the computer used or the relevant measures of problem size. All these
are typical aspects of a user manual.

I don't think that a reader without in depth prior knowledge could
conclude from the paper how exactly the different measures are defined
or computed. The paper does not tell us how these measures are
interpreded or which properties they have e.g. whether they are
intensive or extensive quantities with respect to taking larger
observational windows. The paper shows how results are computed, but
does not show how they are interpreted. This again is quite usual for
software manuals, but not for research articles or case studies.

As a user manual for a software, it is well written, informative,
detailed and well structured. I just added a few comments in the
details section below, where I think it could be improved. The
software and a detailed description is also probably relevant to the
readership of entropy and the readership of entropy is probably very
happy with the level of detail provided in the manual. The open access
character of Entropy is also very adequat for this publication.

Thus in conclusion, it is in essence an editorial decision: If Entropy
sees itself as publishing software manuals, this a highly relevant and
well written contribution, that could be accepted as is or with minor
revisions.

If however Entropy sees itself not as publishing manuals, as I
would read the aims and scope, the paper is simply outside the scope.

And for this reason and only this reason I review it as "reject".

Specific
========

line 291: This explanation of the idea is to unspecific to be a clear definition.

line 300: This advice to rescale is to generic. How do I rescale. How do I interrpret a scaled result. Please be more specific in the advise.

line 403: "require no time and return results instantly": this is
clearly objectively false as any computation costs time. Please
reformualte correctly.

line 522: "few minutes at the longest" This is not a scientific claim without stating the computing system. Please specify.

Round 2

Reviewer 1 Report

Comments and Suggestions for Authors

The quality of the paper has been improved after revision. The author strived to address the issues raised by me. To my thinking, the paper is acceptable for publication.

Comments on the Quality of English Language

Minor editing of English language required

Reviewer 2 Report

Comments and Suggestions for Authors

The main point of critic in my previous review, was the character of
the paper as a software manual rather than a scientific paper. The
authors have fixed this issue perfectly and developed a very well
written tutorial in the form of a case study with review paper
characteristics. They now deliver a highly relevant exposition of the
capabilities of the package using a relevant case study and clear
explanations of the tasks and interpretability of the results. The
function of the paper to equiped a reader with the ability to actually
do such an analysis practically with the package has been maintained,
by shifting the required steps into an appendix. The topic of the
paper is still highly relevant for the readership of Entropy, and the
authors again were able to explain everything very clearly.

In this form the paper is highly beneficial to the readership of Entropy and should be published as is.

I just came across one typo:

Table 7: the last lines give the time only as a unit "mins" and
without a number. Please fix. Probably this can be done in the proofs.